# Recycled iron fuels new production in the eastern equatorial Pacific Ocean

Patrick A. Rafter [1], Daniel M. Sigman[2] & Katherine R.M. Mackey[1]

Nitrate persists in eastern equatorial Pacific surface waters because phytoplankton growth fueled by nitrate (new production) is limited by iron. Nitrate isotope measurements provide a new constraint on the controls of surface nitrate concentration in this region and allow us to quantify the degree and temporal variability of nitrate consumption. Here we show that nitrate consumption in these waters cannot be fueled solely by the external supply of iron to these waters, which occurs by upwelling and dust deposition. Rather, a substantial fraction of nitrate consumption must be supported by the recycling of iron within surface waters. Given plausible iron recycling rates, seasonal variability in nitrate concentration on and off the equator can be explained by upwelling rate, with slower upwelling allowing for more cycles of iron regeneration and uptake. The efficiency of iron recycling in the equatorial Pacific implies the evolution of ecosystem-level mechanisms for retaining iron in surface ocean settings where it limits productivity.

[1] Department of Earth System Science, University of California, Irvine, CA 92697, USA. [2] Department of Geosciences, Princeton University, Princeton, NJ 08540, USA. Correspondence and requests for materials should be addressed to P.A.R. (email: prafter@uci.edu)

The biological fixation and export of carbon to the deep sea—the biological carbon pump—plays an important role in the air-sea partitioning of $CO_2$ on a variety of time-scales[1]. The prevailing paradigm is that the biological pump is driven by photosynthesis fueled by the consumption of new nutrients supplied to the euphotic zone, or new primary production[2], which has shaped the parameterization of carbon in marine biogeochemical models. For example, where phytoplankton growth is limited by the trace metal micronutrient iron, it is widely held that an increase in the iron-to-nitrate supply ratio is necessary to increase the degree of nitrate consumption in surface waters and therefore the biologically driven storage of carbon in the deep sea[3]. However, recent work suggests that this paradigm may be incomplete for iron because we now know that the supply, retention, speciation, and regeneration of iron work together to influence phytoplankton growth and possibly nitrate consumption in high nutrient low chlorophyll (HNLC) regions[4–8]. Hence, new production may be fueled simultaneously by new nitrate and recycled iron, with implications for temporal variability in deep ocean carbon storage.

To understand the controls on the degree of nitrate consumption with respect to iron supply and recycling in HNLC regions, we use the equatorial Pacific upwelling system as a natural laboratory that is free of potential light limitation and is well studied with respect to seawater iron chemistry[4, 9–13] to answer a simple yet fundamental question: can the available iron supply explain the observed nitrate consumption as waters upwell to the surface along the equator and flow poleward? Our findings indicate that the supply of new iron (which is delivered mostly by upwelling and dust deposition) cannot explain the observed nitrate consumption in these iron-limited waters, requiring that iron recycled within the euphotic zone fuels a substantial portion of the nitrate consumption. Iron recycling also provides an explanation for nitrate concentration variability associated with upwelling strength[14]: slower upwelling allows for more cycles of iron reuse, resulting in more complete nitrate consumption and lower surface nitrate concentrations.

## Results

**The subsurface source of equatorial Pacific surface nitrate.** The tropical tradewinds produce a net westward surface circulation at the equator, but the surface pathway of a particular parcel of water at the equator will include both westward and poleward velocities (at a rate of about 1° of latitude per 10 days)[15]. The surface water divergence caused by this poleward circulation drives equatorial upwelling and is strongest in boreal fall when tradewinds increase. Equatorial surface water nitrate concentrations are highest during the fall period of strong upwelling (Fig. 1), but this relationship has not yet been explained in the

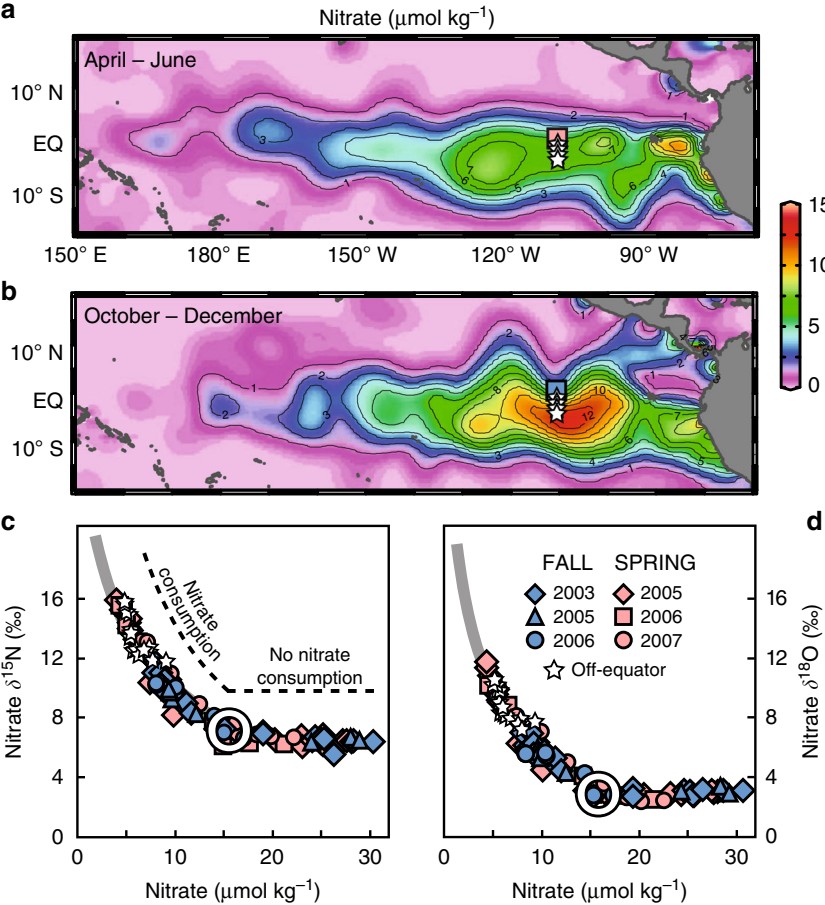

**Fig. 1** Tropical Pacific nitrate concentration and isotopic composition. Tropical Pacific surface nitrate concentrations for **a** April–June and **b** October–December[70], with squares and stars indicating station locations. **c**, **d** show nitrate $\delta^{15}N$ and $\delta^{18}O$ measurements vs. nitrate concentration for the samples from these stations. Pink and blue symbols are ±1° of equator during boreal spring and fall, respectively. White stars indicate 2–4° S surface mixed layer measurements. Plotted data are from the upper 200 m of the water column. White circles in **c**, **d** indicate the averages of measurements of the Equatorial Under Current (EUC) at 110° W[18], and these values are used to drive the Rayleigh (closed system) model of nitrate assimilation (gray lines)[18] (see text for more details)

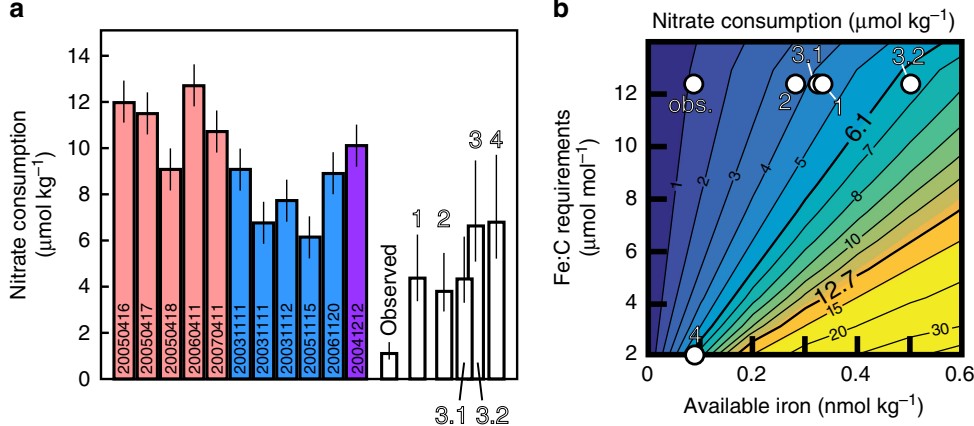

**Fig. 2** Observed and potential nitrate consumption. **a** Nitrate consumption estimates from 2003 to 2007 (colored bars) compared with potential nitrate consumption calculated for Scenarios 1–4 (open bars). Nitrate consumption estimates in **a** are from spring (pink), fall (blue), winter (purple), and error bars are ±1 μmol kg⁻¹ based on source water variability (sampling date at bottom; values in Tables 1 and 2). White columns represent nitrate consumption predicted based on the following scenarios. "Observed" uses iron supply and diatom Fe:C requirements that match local observations[12, 22]. Scenario 1 uses local observations, but with a doubled dust-borne iron supply and an order of magnitude higher solubility of dust-borne iron. Scenario 2 assumes all particulate iron is bioavailable. Scenarios 3.1 and 3.2 use dissolved iron concentrations from 200 m and EUC measurements 3300 km west, respectively. Scenario 4 uses the lowest reported diatom Fe:C requirement of 2 μmol mol⁻¹ [7]. Error bars (1 standard deviation) allow for potential C:N variability of ±2 (global community value[71]) even though phytoplankton community C:N ratio in the eastern equatorial Pacific is much smaller (±0.3[10]). **b** A sensitivity analysis of potential nitrate consumption based on available iron concentrations and physiological iron requirements. White numbers in **b** indicate the Scenarios from **a**, none of which can explain the observed range in nitrate consumption of 6.1 and 12.7 μmol kg⁻¹ (bold contours)

context of iron limitation. The poleward movement of equatorial surface waters limits the westward transport of these waters along the equator, indicating that water and nutrients upwelled at the equator must come from a relatively local subsurface source. This source was previously identified as the local Equatorial Under Current (EUC)[16], an eastward-flowing subsurface equatorial jet positioned within the thermocline.

To test whether the EUC is the source of equatorial surface nutrients and to identify the physical and biological processes affecting seasonal to interannual surface nitrate concentration variability in the region (Fig. 1), we use 5 years of nitrate concentration and isotope ($\delta^{15}$N and $\delta^{18}$O) data from 0° N, 110° W (squares in Fig. 1). Nitrate $\delta^{15}$N and $\delta^{18}$O are elevated equally during nitrate consumption[17], and both are consistently higher during boreal spring when surface nitrate concentrations are lowest (Fig. 1). In the case where nitrate consumption acts on a finite nitrate pool (a closed system), the nitrate isotopes follow the Rayleigh model[18]:

$$\delta^{15}\text{N} = \delta^{15}\text{N}_{\text{initial}} - \varepsilon \times \ln(f) \qquad (1)$$

Where $f$ represents the observed nitrate concentration divided by the initial nitrate concentration and $\varepsilon$ is the isotope effect, estimated as $6.0 \pm 0.5‰$ in this region[14]. Initiating the Rayleigh model using EUC measurements (nitrate concentration of $15.8 \pm 2.2$ μmol kg⁻¹, nitrate $\delta^{15}$N and $\delta^{18}$O of $7.1 \pm 0.2‰$ and $3.2 \pm 0.3‰$[14]) gives the gray line in Fig. 1c, d. Comparing the data with the model, we find that both nitrate $\delta^{15}$N and $\delta^{18}$O in upwelled equatorial waters adhere to the Rayleigh model ($R^2 = 0.94$). Both spring and fall data fit a single Rayleigh trend beginning at a nitrate concentration and isotopic value equivalent to those of the EUC, and this consistency also applies to near-equatorial surface waters that have flowed poleward after Ekman upwelling ($R^2 = 0.93$) (up to 4° south of the equator; stars in Fig. 1). The $\delta^{15}$N and $\delta^{18}$O data call for the same isotope effect for nitrate consumption, consistent with assimilation by phytoplankton being the dominant process affecting euphotic zone nitrate in the region[14].

(Surface nitrate at higher latitudes than 4° S along 110° W is complicated by westward transport from upwelling further east[14].) To allow for subtle temporal variability of source water nitrate concentrations, the initial nitrate concentration in the Rayleigh equation (Eq. 1) was calculated for each station occupation[14], producing data-model correlations of 0.88–0.99 (see Rafter and Sigman[14] Table 1). The average of this Rayleigh-based estimate for subsurface source water nitrate concentration is $16.1 \pm 1.0$ μmol kg⁻¹, nearly identical to the observed EUC nitrate concentration of $15.8 \pm 2.2$ μmol kg⁻¹ at 110° W.

The close correspondence between the nitrate isotopes and a closed system nitrate consumption model has several implications. First, local EUC water must be the dominant source of upwelled nitrate in the region[16] in seasons of both weak and strong upwelling (spring and fall, respectively) and despite variability in the depth of the EUC[19, 20]. Moreover, variability of the EUC (source water) nitrate concentration and isotopic composition must be minor. Second, the decline in nitrate concentration as water upwells from the subsurface to the surface and then moves off-equator must be predominantly caused by in situ nitrate assimilation, as opposed to mixing with off-equatorial surface waters of lower nitrate concentration; this also rules out significant inputs of iron from off-axis waters. Third, assimilation must be the dominant biological flux of nitrate in the surface mixed layer; if nitrification was producing a significant amount of nitrate, we would not observe the parallel elevation of nitrate $\delta^{15}$N and $\delta^{18}$O (Fig. 1)[17].

These cumulative closed system characteristics yield the simple but powerful implication that we can derive the euphotic zone nitrate consumption from the difference in nitrate concentration between local EUC water and the surface mixed layer. We plot the total nitrate consumption for each station occupation in Fig. 2a and Table 1. The results show that the observed—but unexplained—seasonality in surface nitrate concentrations (Fig. 1) derives from changes in nitrate consumption; the degree of consumption is greatest and thus lowers surface nitrate concentration most in boreal spring, when upwelling is weakest (Figs. 1, 2).

**Table 1 Data and calculations for all station occupations**

| Date (YYYY/MM/DD) | Estimated source nitrate concentration (µmol kg⁻¹) | Mixed layer nitrate (µmol kg⁻¹) | Nitrate consumption (µmol kg⁻¹) | Necessary dissolved iron concentration for nitrate consumption (nmol kg⁻¹) |
|---|---|---|---|---|
| 2005/04/16 | 16.0 | 4.0 | 12.0 | 0.98 |
| 2005/04/17 | 16.6 | 5.1 | 11.5 | 0.94 |
| 2005/04/18 | 14.4 | 5.3 | 9.1 | 0.74 |
| 2006/04/11 | 16.9 | 4.2 | 12.7 | 1.04 |
| 2007/04/11 | 17.8 | 7.1 | 10.7 | 0.87 |
| 2003/10/11 | 16.7 | 7.6 | 9.1 | 0.74 |
| 2003/10/11 | 15.0 | 8.3 | 6.8 | 0.55 |
| 2003/10/12 | 15.4 | 7.7 | 7.7 | 0.63 |
| 2005/11/15 | 15.5 | 9.3 | 6.1 | 0.50 |
| 2006/11/20 | 16.9 | 8.0 | 8.9 | 0.73 |
| 2004/12/12 | 15.3 | 5.2 | 10.1 | 0.82 |

Source water nitrate concentration was estimated using nitrate isotope measurements for the first 10 stations in the list. Subsurface source water nitrate concentration for the last station was identified as a subsurface eastward velocity maximum, which is the Equatorial Under Current (based on refs. [14, 16] and see "Methods"). Nitrate consumption is calculated as the difference between the subsurface source water and surface mixed layer nitrate concentration. The dissolved iron concentration required to drive this nitrate consumption was calculated assuming an Fe:C requirement of 12.3[22] and a C:N requirement of 106:16.

**Table 2 Potential nitrate consumption was calculated for a range of iron concentrations and Fe:C physiological requirements**

| | Available iron (nmol kg⁻¹) | Fe:C physiological requirements (µmol mol⁻¹) | Potential nitrate consumption (µmol kg⁻¹) | Note | Reference(s) |
|---|---|---|---|---|---|
| Observed | 0.09 | 12.3 | 1.10 | Observations from 0° N, 110° W | Twining et al.[22]; Kaupp et al.[12] |
| 1 | 0.36 | 12.3 | 4.39 | Double-dust flux and 10 times iron solubility | Winckler et al.[57] |
| 2 | 0.31 | 12.3 | 3.80 | All particulate iron is bioavailable | Gordon et al.[30] |
| 3.1 | 0.35 | 12.3 | 4.30 | Dissolved iron concentrations from 200 m at 110° W | Kaupp et al.[12] |
| 3.2 | 0.54 | 12.3 | 6.63 | Dissolved iron from 0° N, 140° W at 120 m | Kaupp et al.[12] |
| 4 | 0.09 | 2.0 | 6.79 | Very low Fe:C physiological requirements | King et al.[33] |

These nitrate consumption estimates were converted to Fe:N assuming Redfield C:N of 106:16 (see text for more details). "Observed" uses observed values at 0° N, 110° W[12, 22]. Scenario 1 doubles the dust-borne iron supply (equal to last glacial maximum values[57]) with an order of magnitude higher iron solubility (60%). Scenario 2 assumes all eastern equatorial Pacific euphotic zone particulate iron is bioavailable (observations from ref. [30]) and Scenarios 3.1 and 3.2 assume higher dissolved iron concentrations. Scenario 4 uses the lowest reported Fe:C of 2 µmol:mol[33]. Note that the annual range in observed nitrate consumption is 6.1–12.7 µmol kg⁻¹, but the highest calculated value is only 6.79.

**Quantifying the iron required for nitrate consumption.** The nitrate consumption calculations in Fig. 2 raise the question, is the observed consumption of newly upwelled nitrate in these iron-limited waters explained by the delivery of iron? Observations at our study site (0° N, 110° W) indicate that diatoms are the main consumer of equatorial Pacific nitrate[21], and diatoms appear to have a physiological iron:carbon (Fe:C) requirement near 12.3 µmol mol⁻¹ [22]. Assuming a biomass C:N of 106:16[23], the observed range in nitrate consumption (6.1–12.7 µmol kg⁻¹) requires 0.50–1.04 nmol kg⁻¹ of iron. In contrast, source water dissolved iron concentration is 0.09 nmol kg⁻¹ [12] (with an iron-to-nitrate ratio of $6 \times 10^{-6}$), too low by 5–10 fold. Given the above stoichiometry, the subsurface dissolved iron concentration predicts only ≈1.1 µmol kg⁻¹ of potential nitrate consumption (observed in Fig. 2a and Table 2). While the simple assumptions behind these stoichiometric calculations are similar to earlier work[24], they do not take into consideration the iron demand from other phytoplankton, which account for roughly ≈80–90% of the phytoplankton biomass[25]. This calculation therefore provides an upper bound for the potential nitrate consumption.

**Sensitivity tests of potential nitrate consumption.** The missing supply of iron cannot be explained by atmospheric deposition, which has been measured at ≈0.01 µmol Fe m⁻² day⁻¹, or <2% of

the upwelling iron flux (0.55 µmol Fe m⁻² day⁻¹) at this site[12, 24]. These dust-borne iron flux measurements compare well to average dust flux rates over the Holocene estimated from sediment underneath our study site (0.02 µmol Fe m⁻² day⁻¹; see Methods section), suggesting that these very low fluxes are reasonable. Regardless, we can identify the sensitivity of our potential nitrate consumption calculation by considering the possibility of higher dust-borne iron fluxes. For example, in Scenario 1 (Fig. 2a and Table 2), the solubility of dust-borne iron is increased by an order of magnitude while the dust-borne iron supply is doubled to last glacial maximum (LGM) values. This Scenario 1 predicts a potential nitrate consumption of only 4.4 µmol kg⁻¹ (see "Methods"). Thus, even under these unlikely conditions, nitrate consumption is still far too low to explain the observed range in nitrate consumption (Table 2). This calculation is consistent with nitrogen isotope records indicating that dust-borne iron had a limited influence on equatorial Pacific nitrate consumption during the LGM[26, 27]. Furthermore, considering that global atmospheric dust concentrations during the LGM were the highest of the past 4 million years[28], it is likely that dust has played a limited role in alleviating iron-limiting conditions in the eastern equatorial Pacific over this entire period.

Scenario 2 represents a case where particulate iron is partially bioavailable[29] (Fig. 2 and Table 2). Particulate iron concentrations are ≈0.3 nmol kg⁻¹ in the upper eastern equatorial Pacific[30],

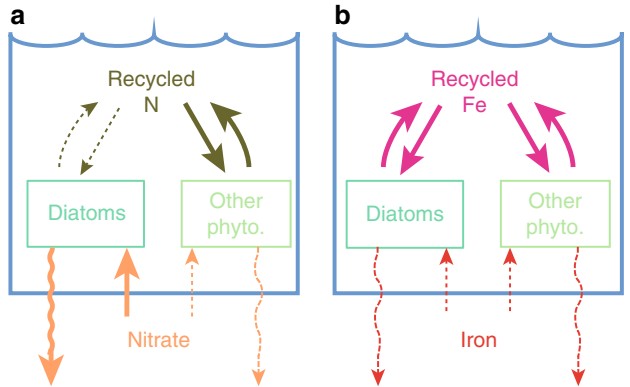

**Fig. 3** Distinct iron and nitrogen cycling pathways and their affect on nitrate consumption. Conceptual models of **a** nitrogen (N) and **b** iron (Fe) transformation pathways in the eastern equatorial Pacific, where nutrient consumption is shown by a straight arrow, nutrient regeneration by a curved arrow, and wavy arrows denote export. Arrow size denotes relative flux size, and dashed arrows are very small fluxes (see text for details). Other phyto. refers to non-diatom phytoplankton, including autotrophic flagellates and picoplankton[25], which primarily meet their N requirements with recycled N[21] (see "Methods"). The preferential regeneration of iron relative to nitrogen predicts a higher export of nitrogen relative to iron

and we make the extreme assumption that all of this particulate iron as well as the measured dissolved iron of 0.09 nmol kg$^{-1}$ is available to nitrate-consuming phytoplankton. However, Scenario 2 predicts nitrate consumption of only 3.8 µmol kg$^{-1}$. Higher particulate iron concentrations are observed deep below the central equatorial Pacific EUC[31], but these waters are too deep and too far west to be upwelled to the surface at our study site[14].

Scenarios 3.1 and 3.2, respectively, assume the upwelling of higher dissolved iron concentrations from deeper water (200 m) or EUC water from further west (at 140° W). In Scenario 3.1, the potential nitrate consumption is 4.3 µmol kg$^{-1}$, still failing to match the observations (Table 2). The deeper waters considered in Scenario 3.1 also have much higher nitrate concentrations, which would be evident as an offset in the nitrate δ$^{15}$N vs. nitrate concentration relationship to the right (higher nitrate concentrations) in Fig. 1c, d, but is not observed at any station occupation. Scenario 3.2 yields nitrate consumption of 6.6 µmol kg$^{-1}$ (Fig. 2 and Table 2). While this is close to the lowest observed nitrate consumption, it cannot explain the full range (Table 1). There are also several lines of evidence suggesting that variability of EUC dissolved iron concentrations in the EUC is small and that the higher iron concentrations to the west are not transmitted to the eastern equatorial Pacific[12, 32] (see "Methods").

Scenario 4, a different phytoplankton stoichiometry, is also unable to explain nitrate consumption. Our standard calculation assumes an Fe:C requirement of 12.3 µmoles mol$^{-1}$ based on synchrotron x-ray fluorescence measurements of diatoms at 0° N, 110° W in December 2004[22], which are consistent with similar measurements from the South Pacific[33]. Different methodologies predict both higher and lower Fe:C requirements[33], and Scenario 4 assumes the lowest reported Fe:C requirement of 2 µmol mol$^{-1}$ (based on a laboratory culture[7]). Still, it cannot explain the observed nitrate consumption in the eastern equatorial Pacific (Fig. 2 and Table 2). Moreover, dinoflagellates may also consume nitrate at this site[21], and they appear to have a higher Fe:C requirement (14.2 µmol mol$^{-1}$[22]), only serving to exacerbate the mismatch. More efficient iron consumption (a lower Fe:C) has been observed as phytoplankton communities evolve during upwelling[34], but this may not apply to the eastern equatorial Pacific phytoplankton community, which has been described as

having "remarkable constancy"[25]. In any case, such community changes are also unlikely to drive the Fe:C requirement below the extremely low diatom Fe:C requirement of 2 µmol mol$^{-1}$ of Scenario 4. A higher iron requirement typically applies at low-light levels[35], but we did not include this effect in our calculations, as it would only reduce the calculated potential nitrate consumption.

Despite the extreme assumptions of Scenarios 1–4, the nitrate consumption for each is lower than all but the lowest observed amplitude of nitrate consumption (Fig. 2). Some process must cause greater nitrate consumption than can be supported by the gross rate of iron supply to the surface mixed layer (Table 2).

## Discussion

Internal cycling of iron within the euphotic zone (iron recycling) is the only remaining process capable of explaining the higher-than-expected nitrate consumption[7]. Iron recycling in equatorial Pacific surface waters has been directly observed[4] and is necessary to explain gross primary production rates[5]. Furthermore, both prokaryotic and eukaryotic phytoplankton have been observed to consume recycled iron in HNLC regions[36]. In the conceptual picture of surface ocean elemental cycling that arises, N recycling is most active among the smaller phytoplankton such as *Prochlorococcus* and photosynthetic flagellates (consistent with refs. [21, 37]), while recycled iron emanates from and is in turn available to all biota (Fig. 3). The key parameter in the model required for iron recycling to fuel nitrate drawdown is a Fe:N remineralization ratio that is higher than the data-constrained plankton biomass Fe:N ratio of 12.3 µmol mol$^{-1}$ [22].

To estimate the necessary biological rates to simulate surface nitrate data, we constructed a numerical model of the nutrient transformations that occur as local EUC waters are upwelled to the equatorial surface and are then advected away from the equator (see "Methods"). This simple biogeochemical model reproduces the drawdown of nitrate (Fig. 4) with a nitrate uptake rate (1–5 nmol kg$^{-1}$ h$^{-1}$) and phytoplankton growth rates (≈2 day$^{-1}$ for diatoms and ≈0.8 day$^{-1}$ for picoplankton and flagellates) similar to observations[21, 38]. Likewise, optimized model settings produced phytoplankton grazing and loss rates that were nearly equal to the average growth rate for each phytoplankton group, consistent with the "balance hypothesis" of Landry et al.[39]

While the consistency between these observed and modeled processes suggests that the model captures the dominant processes that control iron availability in this region, it does not simulate all processes that could potentially influence the cycling of iron and nitrogen. For example, we did not simulate the uptake of silicic acid, which may affect nitrate uptake indirectly[40]. The model neglects iron loss via scavenging because, in this region, scavenging rates are orders of magnitude slower than biological uptake rates and hence have little influence on free iron (Fe′) availability (see calculations in "Methods"). We also do not explicitly model the many processes that are likely involved in extending the residence time of iron in the euphotic zone, including iron-binding ligands, the photoreduction of organic iron complexes in the surface layer[41], viral lysis[36], and heterotrophic grazing, the last of which appears to both increase the solubility of digested iron[42] and release additional iron-binding ligands[43]. While our model does not simulate these processes explicitly, the iron uptake parameters used in the model are based on observations and therefore implicitly include their influence.

Modeled nitrate consumption provides a timescale for the observed range of nitrate consumption of 6.1 µmol kg$^{-1}$ in 125 days, 12.7 µmol kg$^{-1}$ after 260 days, and complete consumption in ≈300 days (Fig. 4). This timescale is consistent with observed nitrate uptake rates as waters upwell to the surface

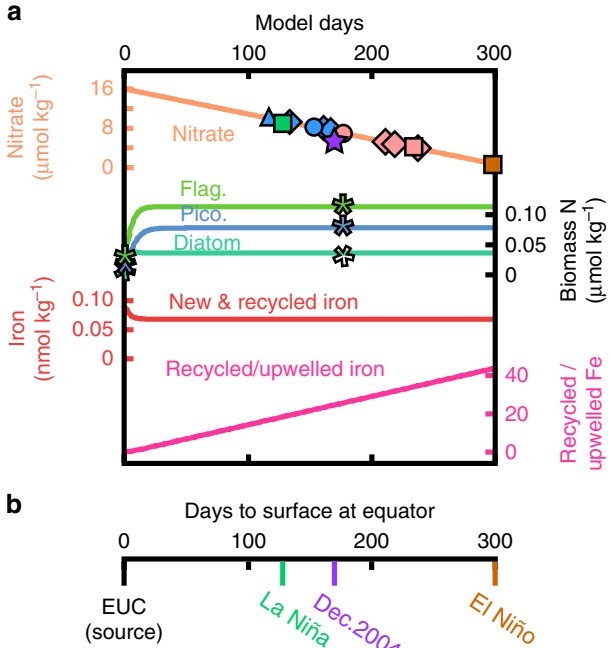

**a**

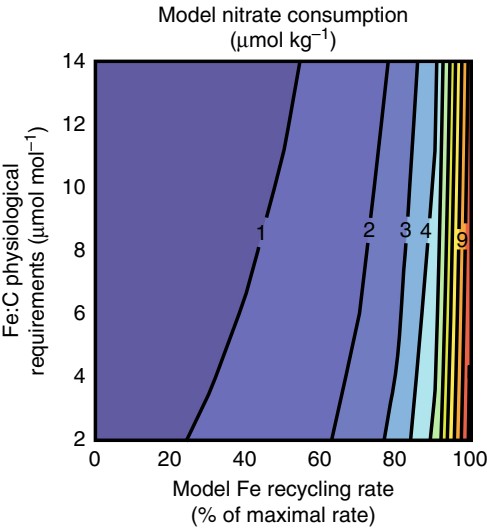

**Fig. 5** Sensitivity analysis of model nitrate consumption. Modeled nitrate consumption at 200 model days as a function iron recycling rate (X-axis) and iron: carbon (Fe:C) physiological requirement (Y-axis). With no iron recycling and a Fe:C requirement of 12.3, the model predicts nearly identical nitrate consumption as the stoichiometric calculation using observed values (0.9 vs. 1.1 μmol kg$^{-1}$; Supplementary Table 1). Increasing model recycling and/or decreasing the Fe:C physiological requirement increases the predicted nitrate consumption. The maximal iron recycling rate considered is based on optimized model settings (see "Methods")

**Fig. 4** Biogeochemical model of upwelled equatorial water. **a** The output of a numerical box model following the nutrient transformations shown in Fig. 3 shows the observed decline in nitrate and increase in phytoplankton biomass nitrogen (N) at our study site[14, 25]. Model dissolved iron concentrations (Fe) include both upwelled new and recycled iron and therefore do not drop to the lowest observed regional surface ocean values (≈0.02 nmol kg$^{-1}$)[11, 12]. The ratio of recycled to upwelled iron indicates the degree of iron recycling required to drive the nitrate consumption. Symbols plotted over the modeled nitrate concentration indicate surface nitrate concentration observations from the seasonal stations (orange and purple symbols, Fig. 1), from December 2004 (purple star)[12], and for La Niña (green square) and El Niño conditions (brown square) (based on ref. [19]). Asterisks mark biomass N targets for model tuning at 0.1% and 100% light levels for photosynthetic flagellates (flag.), picoplankton (pico.), and diatoms[25]. **b** The rate at which waters upwell from the Equatorial Under Current (EUC) and advect away from the equator affects its residence time at the equator (how long water takes to upwell to the surface). Here we represent this variable time scale based on surface nitrate concentrations for the December 2004 measurements (purple) and during La Niña and El Niño conditions[19]. Poleward advection of these waters once they reach the surface should scale accordingly, explaining the observed changes in the spatial extent of HNLC waters

(e.g., refs. [21, 44]) and the approximate time for complete nitrate consumption as upwelled waters reach the surface and are then advected to the edge of the surface nitrate pool (8–10° S). (nitrate in water advected north of the equator is subducted at ≈2° N[45]).

Varying the availability of recycled iron in the model yields predictable results (Fig. 5). In particular, when recycled iron is unavailable (iron recycling is turned off in the model), we find minimal nitrate consumption (<1 μmol kg$^{-1}$). This value is lower than the estimated nitrate consumption from observations (Table 2) because the numerical model takes into consideration iron requirements of non-nitrate-consuming phytoplankton.

To match the observed range of nitrate consumption (Fig. 2 and Table 2), phytoplankton in the model require 18–38 times more iron than would be supplied by upwelling alone, an iron recycling rate of 26 pmol kg$^{-1}$ day$^{-1}$. This rate is consistent with observations near New Zealand[7], although this comparison is complicated by differences in temperature and other conditions in the two regions (see below).

The efficiency of nitrogen and iron recycling can be quantified in terms of the uptake ratio of the new (i.e., upwelled) nutrient to new-plus-recycled nutrient: the *f*-ratio for nitrogen uptake and *fe*-ratio for iron uptake[46]. The *f*-ratio produced by the model (0.34) agrees well with the experimentally observed value for the eastern equatorial Pacific (0.39)[21]. The modeled eastern equatorial Pacific *fe*-ratio ranges from 0.03 to 0.05, which is significantly lower than previous estimates of 0.06–0.51 off of New Zealand[46]. It is possible that physicochemical differences are responsible for these different *fe*-ratios. For example, higher temperatures could elevate biological iron recycling rates at the equator, while increased equatorial irradiance could also amplify the photoreduction of iron–ligand complexes and/or the photochemical cycling of iron. Additionally, there are fundamental differences in high- and low-latitude Pacific biological communities, most notably a much larger presence of cyanobacteria at the equator[38]. Similarly, it has been argued that the highly iron-efficient eastern equatorial Pacific community persists because iron is initially introduced to the system at depth (via upwelling), where more iron-efficient, low-light phytoplankton species can thrive[38]. All of these factors would serve to reduce the *fe*-ratio in this region compared to waters off New Zealand.

More broadly, we are compelled by the possibility that organisms and ecosystems adapt to more efficiently use their limiting nutrient, which is iron in the equatorial Pacific. The finding that the *fe*-ratio is lower than the *f*-ratio in the equatorial Pacific indicates that iron is recycled more efficiently than N, which is a necessary condition for iron recycling to enhance nitrate drawdown. The greater efficiency of iron recycling in this iron-limited region is consistent with the long-recognized but poorly understood paradigm that ecosystem-wide recycling of a nutrient is more intense, where that nutrient is limiting (e.g., ref. [47]). Heterotrophs play a crucial role in iron recycling[42], and efficient iron recycling may be driven by the high-iron demands of heterotrophs themselves[46, 48]. It is also possible that organisms

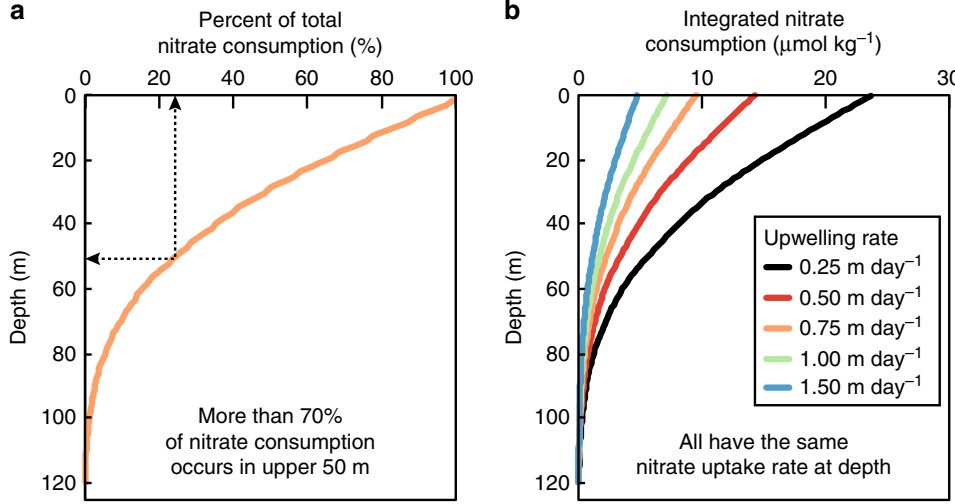

**Fig. 6** Simple models of nitrate consumption at the equator. Nitrate consumption increases as equatorial waters upwell towards the surface, and the degree of consumption is related to upwelling stength. Here we plot the nitrate consumption **a** as a percentage of the initial nitrate concentration and **b** in terms of nitrate concentration with varying upwelling rates. The integrated percentage of nitrate consumption in **a** is based on observed nitrate uptake rate with depth[21] and total nitrate consumption (Table 1) at 0° N, 110° W from December 2004. Note that most nitrate consumption occurs in the upper 50 m, suggesting that light limitation can only play a minor role in modifying elemental requirements (e.g., lower C:N at lower light levels[71]). The same nitrate uptake rates were used to calculate the integrated nitrate consumption in **b** for a variety of upwelling rates, and we found that increasing upwelling rate decreases the integrated nitrate consumption in the water upwelling at the equator (see "Methods"). The same source water depth and iron recycling rate are assumed in all calculations

in the community have co-evolved to create an iron retaining system[6], yielding a low *fe*-ratio in iron-poor systems[49].

From an observational perspective, an *fe*-ratio that is less than the *f*-ratio predicts that the Fe:N of the export flux from the equatorial Pacific euphotic zone is lower than the Fe:N of euphotic zone diatom biomass. Sinking diatoms appear to export iron preferentially relative to phosphorus in the high-iron waters off of New Zealand[50, 51], but this observation is likely influenced by substantial iron scavenging in the shallow subsurface, and it is occurring in the context of a relatively iron-rich system. In general, if iron-poor ecosystems evolve to retain iron, we would expect the Fe:N of many components of the sinking flux to be lower in the equatorial Pacific than in New Zealand. Analogous investigations of the Fe:N of sinking material (excluding undissolved aeolian deposition) in the iron-poor waters of the equatorial Pacific should clarify the importance of iron recycling and export under iron limitation.

The temporal variation of nitrate consumption in the eastern equatorial Pacific is not explained by changes in the iron-to-nitrate supply ratio, which has minor seasonal variability (see above, refs. [9, 12, 30], and "Methods"). Instead, it appears coupled to circulation-driven nitrate supply rates, with weaker upwelling corresponding to higher nitrate consumption and lower surface nitrate concentrations throughout the eastern equatorial Pacific (Fig. 1)[19, 21, 22]. This apparent relationship between upper ocean physics and nitrate consumption can be explained by iron recycling. The weaker upwelling during boreal spring and El Niño events increases the residence time of upwelled waters, allowing more cycles of iron uptake and release and therefore more nitrate consumption as waters upwell to the surface and are advected away from the equator. This is demonstrated in Fig. 6, where surface nitrate concentrations vary according to the rate of upwelling (using observed nitrate consumption rates[21], which we have argued are fueled by recycled iron). The axis in Fig. 4b uses the model output to illustrate the necessary time for nitrate consumption as waters upwell to the surface during the December 2004 cruise (Table 1) and under La Niña and El Niño conditions, based on observations and assuming the parameters from

ref. [19]. The poleward decline in surface nitrate concentration is similarly affected by iron recycling. With slower upwelling and off-equatorial surface water advection during El Niño events, more cycles of iron regeneration and uptake will have occurred at a given distance from the site of upwelling, causing the region of high surface nitrate concentration to contract toward the equator (Figs. 2 and 4b).

Given the potential for iron recycling to mediate nitrate consumption in this and other iron-limited regions, the processes described here may help to explain past changes in the degree of nitrate consumption. Data from the Antarctic indicate an ice age increase in nitrate consumption that was strongly coupled to reduced deep-surface exchange but that pre-dated the rise in dust-borne iron[52]. This increase in the degree of nitrate consumption was potentially important for lowering atmospheric $CO_2$[53], but it has been a challenge to explain why the degree of nitrate consumption was so strongly (inversely) related to deep-surface exchange. Iron recycling within the Southern Ocean euphotic zone provides a simple explanation for this observation: a reduction in deep-surface exchange increased the residence time of surface mixed layer water, allowing iron recycling to fuel more complete nitrate consumption.

In summary, our study identifies the subsurface source of waters upwelling to the eastern equatorial Pacific surface and uses this information to quantify surface nitrate consumption over the course of 5 years. We find that the temporal variability of nitrate consumption in this iron-limited system is closely correlated with upwelling rates and cannot be explained by the available iron supply, regardless of the iron source. These results suggest that internal iron cycling in this HNLC region leads to much more nitrate consumption than would be fueled by new iron inputs alone. Rates of iron cycling required to explain observed nitrate consumption are consistent with observations. The (inverse) correlation of nitrate consumption with upwelling rate can also be explained by iron recycling: slower upwelling increases the residence time of equatorial Pacific surface water, which increases the number of cycles of iron use since the water was upwelled, increasing nitrate consumption and thus lowering nitrate

concentration at a given location in the equatorial Pacific surface. Finally, these findings provide a possible explanation for observed variability in nitrate consumption on longer time scales, ranging from seasonal to millennial.

## Methods

**Sampling metadata and nitrate isotope measurements.** Nitrate N and O isotope measurements (Fig. 1c, d) were made using the denitrifier method[54, 55] on samples obtained along 110° W in the eastern equatorial Pacific (squares and stars in Fig. 1) from six research cruises on NOAA ships R/V Ka'imimoana and R/V Brown between 2003 and 2007. The N isotopic composition of nitrate is expressed in delta notation, where $\delta^{15}N = (^{15}N/^{14}N_{sample}) / (^{15}N/^{14}N_{reference})-1$, referenced to atmospheric $N_2$ and expressed in per mil (‰) by multiplying by 1000. The O isotopic composition of nitrate ($\delta^{18}O$) follows the same equation, but is referenced to VSMOW. Error bars are smaller than symbols in Fig. 1. All data are available at BCO-DMO.org.

**Calculating source water nitrate concentrations.** We identified the nitrate concentration of upwelling waters by first plotting each station's nitrate isotope data vs. the natural log of nitrate concentration. In this space, the data form a straight line when closed system conditions apply. The slope of this line approximates the isotope effect, and the Y-intercept provides the nitrate concentration preceding nitrate consumption. In these calculations, we assumed the initial nitrate $\delta^{15}N$ and $\delta^{18}O$ equaled the basin-wide values for the EUC, which is supported by several studies[14, 56]. Nitrate isotopes were measured for the first 10 stations listed in Table 1 (2003–2007) and allow for 20 estimates of source water nitrate concentration using the above method (one for each nitrate $\delta^{15}N$ and $\delta^{18}O$ profile). These estimates yield an average eastern equatorial Pacific source water nitrate concentration of $16.1 \pm 1.0$ µmol kg$^{-1}$. This value compares well with simply identifying the subsurface velocity maximum for each cruise (the EUC), which yields a subsurface source nitrate concentration of $15.8 \pm 2.0$ µmol kg$^{-1}$ [14]. Applying the same approach to the last station listed in Table 1 (during the Equatorial Biocomplexity Project in 2004) yields a similar subsurface source nitrate concentration of 15.3 µmol kg$^{-1}$. If our assumption of a closed system is wrong or the supply of nutrients is better represented by an open system model (allowing for resupply of new nutrients[14], which may be the case for one station occupation[14], then the actual initial nitrate concentration would be higher than we assume, leading to an underestimation of nitrate consumption (Table 1). In other words, violating the closed system assumption exacerbates the difference between the potential and estimated nitrate consumption.

**Calculating dust-borne iron supply today and during the LGM.** Dust-borne iron supply to the eastern equatorial Pacific is estimated to be 0.01 µmol Fe m$^{-2}$ day$^{-1}$ [12] or <2% of an upwelling flux of 0.55 µmol m$^{-2}$ day$^{-1}$ [12]. This flux is similar to observations from the central equatorial Pacific of 0.01–0.03 µmol Fe m$^{-2}$ day$^{-1}$ [9], although neither estimate takes into account temporal variability. We can identify the long-term average dust-borne iron flux to our site at 0° N, 110° W by examining the accumulation of dust in deep-sea sediment below our study site. The average dust flux over the past several thousand years was 0.13 g dust m$^{-2}$ year$^{-1}$ [57]. Assuming that this largely South American dust is 7.3% iron oxide[58], has a solubility of 6%[12], and is dissolved instantaneously within the mixed layer, gives average Holocene dust-borne iron flux of 0.02 µmol Fe m$^{-2}$ day$^{-1}$, or essentially the same as observed. Together, these estimates suggest that the upwelling iron flux dominates the modern supply of iron to the eastern equatorial Pacific[12].

The sedimentary record also helps to illustrate the limited impact of dust flux variability on nitrate consumption in this setting. For example, during the LGM (roughly 20,000 years ago), dust flux was more than double (0.3 g dust m$^{-2}$ year$^{-1}$ [57]). Using the parameters above, this flux translates into 0.05 µmol Fe m$^{-2}$ day$^{-1}$ or only ≈9% of the modern dissolved iron upwelling flux. In the Scenario 1 calculation (Table 2 and Fig. 2a), we assume the LGM dust-borne iron supply above, a diatom Fe:C requirement of 12.3 µmol mol$^{-1}$ [22], dust-borne iron solubility of 60%, a mixed layer depth of 19 m (average of all cruises in Table 1), and a residence time of 10 days near the equator (see above). This gives a total additional iron contribution from dust of 0.27 nmol kg$^{-1}$ and an additional nitrate consumption of only 4.4 µmol kg$^{-1}$ (Table 2), which is far too low to explain the observed range in nitrate consumption.

**Equatorial Pacific dissolved iron variability.** Measurements of dissolved iron concentration in the central equatorial Pacific (upstream of eastern EUC source waters) indicate that dissolved iron in the upper 200 m has small seasonal and interannual variability, with maximum dissolved iron concentration differences of 0.04 and 0.12 nmol kg$^{-1}$, respectively (calculated with data from [9]). Iron measurements using a different methodology observed similar seasonal variability of 0.03 nmol kg$^{-1}$ (calculated with data from [12]). Decadal variability of dissolved iron concentrations is potentially obscured by methodological differences, but could be as high as ≈0.2 nmol kg$^{-1}$ [12]. Regardless, these potential changes in dissolved iron concentration are still a factor of 5 too low to explain the observed nitrate consumption (Table 1).

**Productivity feedback lowers dissolved iron concentrations.** Dissolved iron measurements are consistently lower in the eastern equatorial Pacific for two reasons. First, measurements of dissolved iron concentration in the central equatorial Pacific (upstream of eastern EUC source waters) indicate that dissolved iron in the upper 200 m has small seasonal and interannual variability (see above). Second, a productivity-scavenging negative feedback appears to keep eastern equatorial Pacific dissolved iron concentrations persistently low[12, 32]. In this feedback, a transient increase in dissolved iron concentrations in the eastward-flowing EUC first increases primary production in the iron-limited western and central equatorial Pacific[59], increasing productivity and sinking particles, which lowers subsurface dissolved iron concentrations by scavenging/adhesion[12, 32]. This feedback helps explain why dissolved iron concentrations in the eastern equatorial Pacific EUC are consistently higher than in the central EUC[12, 32]. This feedback further works to dampen EUC iron concentration variability downstream in waters that upwell in the eastern equatorial Pacific.

**Calculating integrated nitrate consumption with depth.** As waters upwell to the equatorial surface, nitrate uptake rates increase by several fold (1–5 nmol L$^{-1}$ h$^{-1}$)[21]. Combining this data with observed nitrate consumption from the same sampling period (December 2004 in Table 1), we can identify both the percent of total nitrate consumption (Fig. 6a) and the upwelling rate of these waters (Fig. 6b). The latter was accomplished by identifying the upwelling rate that produces nitrate consumption values (based on observed nitrate uptake rates) that match the estimated nitrate consumption of 10.2 µmol kg$^{-1}$ (Table 1). This upwelling rate was 0.7 m day$^{-1}$, indicating a residence time of 170 days to produce a 10.2 µmol kg$^{-1}$ nitrate consumption, which is consistent with our model output (Fig. 4). We can also use the nitrate uptake rates to identify the most important depths for nitrate consumption. Multiplying these nitrate uptake rates at 1 m intervals with the average estimated upwelling rate of 0.7 m day$^{-1}$, we find that the vast majority of the 10.2 µmol kg$^{-1}$ nitrate consumption (>70%) occurs in the upper 50 m (Fig. 6a).

Applying the same nitrate uptake rate calculation with variable upwelling rates (Fig. 6b) demonstrates the large range in nitrate consumption associated with variable upwelling rates. With these assumptions, slower upwelling produces much larger nitrate consumption, especially near the surface. This does not take into account the typical decrease in upwelling for the upper 25 m at this longitude[44], which would further magnify surface nitrate consumption at the shallowest depths. It is possible that the importance of the upper euphotic zone to total nitrate consumption likely explains why there is no apparent lag between the relatively short-term changes in seasonal upwelling strength and the significantly longer total residence time of nitrate at the equator.

**Numerical model details.** To provide a timescale for our conceptual model of eastern equatorial Pacific nitrate consumption and iron recycling (Fig. 3), we constructed a box model to simulate the upwelling of water to the equatorial surface followed by poleward advection at the surface (Fig. 1). The model was constructed using the Berkeley Madonna (v. 9.0.122) modeling software package that numerically solves ordinary differential and difference equations (http://www.berkeleymadonna.com) (see Supplementary Fig. 1 and Supplementary Note for model parameters).

As observed[14], our numerical model is a closed system and nutrients are not resupplied to the surface (in contrast to "chemostat-like regulation" of nutrients[60]). Dynamic growth rates were determined by Monod nutrient kinetics for nitrogen and iron. The model includes three phytoplankton groups: diatoms whose N requirements are fulfilled by nitrate (based on observations[61]), Prochlorococcus[38] who consume recycled N products[21]; and photosynthetic flagellates (representing autotrophic flagellates and dinoflagellates[25]) who meet their N quota by recycled N[21]. Some dinoflagellates consume nitrate at this site[21], but their contribution to nitrate consumption is poorly constrained and their higher Fe:C requirement[22] would increase the estimated iron recycling rate of the model. Likewise, the bioavailability of recycled iron is unknown and, for the sake of simplicity, is assumed to be available to all phytoplankton groups. There is no nitrification in the model, which is consistent with observations[14].

Diatom and photosynthetic flagellate nutrient requirements are based on eastern equatorial Pacific diatom Fe:C observations[22] converted to Fe:N using a Redfield C:N relationship of 106:16. Prochlorococcus Fe:C requirements are not available, but we assume they are 25% of diatom requirements based on observed Fe:P quotas[22]. This lower iron requirement is consistent with the observation that Prochlorococcus has fewer metalloenzymes[62] and therefore have considerably lower iron requirements[63]. Initial biomass values were based on observations[25] and are converted using a C:N of 106:16. Most remaining diatom and picoplankton parameters are taken from the Community Earth System Model Biogeochemical Elemental Cycling model (CESM)[64] (Supplementary Table 1). Remaining photosynthetic flagellate parameters were assumed to be equal to diatoms.

Grazing is imposed on each phytoplankton group independently, and the model allows iron and nitrogen to be recycled and lost from separate detrital pools. We prescribe a 10% permanent loss of picoplankton and nanoplankton detrital N and a 30% permanent loss of diatom detrital N based on CESM settings[64]. The model was fit to observed biomass values for each phytoplankton group[25] by adjusting the maximum growth rate and grazing pressure (note that grazing also adjusts the N and iron available for recycling).

The model neglects iron export or loss by scavenging because scavenging rates are negligible relative to uptake in this region (see calculations in "Methods"). This assumption therefore provides iron recycling estimates that are conservative, lower-bound values. Likewise, the model does not explicitly incorporate the increased bioavailability of iron caused by organic complexation with ligands. The most complex biogeochemical models have only recently begun to incorporate simplified iron scavenging processes and the organic compounds that protect iron from scavenging (see ref. [11] and "Methods"), therefore our ability to accurately model these processes is still in its infancy[65]. Importantly, the explicit parameterization of iron loss by sinking vs. scavenging and of the bioavailability effects of ligands would not change the core observation of our study, which is that new / upwelled dissolved iron cannot explain the observed nitrate consumption.

**Comparative calculations of iron fluxes**. The concentration of marine dissolved iron includes both the concentration of free iron (Fe′) and that of iron complexed with organic molecules known as ligands (ligand-bound iron). The scavenging of iron onto sinking particles is an important loss term in many parts of the ocean, but ligands protect iron from scavenging. Therefore biogeochemical models that seek to replicate marine iron pool dynamics such as scavenging must include both free- and ligand-bound iron.

Our simple model does not aim to realistically reproduce iron pool dynamics, but we have performed calculations to show that iron scavenging is negligible relative to iron uptake by phytoplankton, supporting the absence of scavenging in our model. Without iron scavenging, there is no need to explicitly model Fe′ concentrations and ligand dynamics. The calculations are outlined below.

The governing equation for total iron (FeT) flux is:

$$\frac{dFeT}{dt} = SFe - kscFe' - \mu_{max}\frac{FeT}{FeT+kFe}BFe$$
$$FeT = Fe' + FeL$$
$$LT = L' + FeL \qquad (2)$$
$$\beta = \frac{FeL}{Fe'L'}$$

Where $S$ is the source flux of iron, ksc is a constant associated with scavenging of Fe′, $\mu_{max}$ is the maximum iron uptake rate, kFe is the uptake rate constant, and BFe is biomass iron. In order to compare the iron scavenging flux (term 2 in Eq. 2) and iron uptake flux (term 3), we must calculate Fe′, which is performed by solving the following equations for total dissolved iron (FeT), total ligands (LT), and the scavenging conditional stability constant ($\beta$). $L'$ represents free ligands and FeL represents ligand-bound iron. Total dissolved iron in surface waters is 0.02 nmol kg$^{-1}$ (based on the minimum observed surface ocean values by ref. [12]), and we use a total ligand concentration of 1 nmol kg$^{-1}$ (observations are between 0.5 and 1.0[11]). The conditional stability constant ($\beta$) is between $1 \times 10^{10}$ and $1 \times 10^{12}$ M$^{-1}$ (see ref. [11] or ref. [66]). Using the smaller $\beta$, the resulting quadratic equation can be solved numerically to give an Fe′ concentration of $2 \times 10^{-12}$ nmol kg$^{-1}$ ($2 \times 10^{-14}$ with the larger $\beta$), much lower than the concentration of total iron. Allowing for colloidal iron to be scavenged and assuming this is $\approx$50% of total dissolved iron (as seen in the North Pacific[67]) gives a much larger value of 0.01 nmol kg$^{-1}$.

Models such as the MITgcm typically use an iron scavenging constant (ksc) of $1 \times 10^{-8}$ s$^{-1}$ [68] (similar to highest observed value from ref. [69]). Using this value and the higher Fe′ concentration above of $2 \times 10^{-12}$ nmol kg$^{-1}$, Eq. 2 yields that the scavenging flux is $2 \times 10^{-20}$ nmol kg$^{-1}$ s$^{-1}$. Assuming colloidal iron is also scavenged increases this scavenging flux to $1.6 \times 10^{-11}$ nmol kg$^{-1}$ s$^{-1}$.

The iron uptake term (final term in Eq. 2) is calculated using typical values from the literature and numerical models[68]: $\mu_{max} = 1 \times 10^{-5}$ s$^{-1}$, kFe = 0.01 nmol kg$^{-1}$, FeT of 0.02 nmol kg$^{-1}$, and BFe = 1 nmol kg$^{-1}$.

This gives an uptake rate of $\approx 7 \times 10^{-5}$ nmol kg$^{-1}$ s$^{-1}$, which is many orders of magnitude larger than either calculated scavenging terms above. Therefore, scavenging is minor relative to iron uptake when considering iron flux in the surface ocean, and we can neglect this term in our simple model. If iron scavenging is minor, then the lack of iron–ligand dynamics in our numerical model is not a substantial concern.

**Data availability**. Data can be found in the BCO-DMO database (https://www.bco-dmo.org/dataset/615082/data).

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

## Acknowledgements

We thank NOAA for sample acquisition and the following people for comments on the manuscript: J. Granger, J. K. Moore, R. Letscher, K. Barbeau, E. Druffel, B. Hopkinson, B. Keller, A. King, D. Marconi, A. Martiny, C. Measures, F. Morel, F. Primeau, K. Selph, E. Sherman, B. Twining, and eight anonymous reviewers. We also sincerely thank M. Follows for providing us with the ligand dynamics calculations and for values from the MITgcm to test model conditions. This work was supported by NSF grant 0960802 to D.M.S. and the Grand Challenges Program of Princeton University.

## Author contributions

P.A.R. generated all data. P.A.R. and K.R.M.M. developed the numerical model. All authors contributed to the concepts and text.

## Additional information

**Competing interests:** The authors have no competing financial interests.

**Change history:** A correction to this article has been published and is linked from the HTML version of this paper.

