## [Peer Review File · Nature Communications]

Reviewers' Comments:

Reviewer #1:

Remarks to the Author:

This paper makes the case that recycled iron supports nitrate-supported new production in the Equatorial Pacific HNLC region. Basic assumptions of this model are that the upwelling process here can be treated as a closed system relative to iron and nitrate supply, as the water is upwelled, reaches the surface and advects poleward. Their calculations suggest that iron must be far more efficiently recycled than nitrogen, relative to phytoplankton demand, in order for the limited new inputs from the original upwelling event to support observed levels of nitrate drawdown. It does seem reasonable that Fe might be much more efficiently conserved in the Equatorial Pacific than N, since Fe is the primary limiting nutrient there. As they note, it is certainly likely that the plankton community here has been selected to hold tightly onto this key limiting resource.

This argument really comes down to a basic textbook concept in oceanography, the f ratio, or the ratio of new production to total production, usually applied to nitrogen. An analogous value for iron is often called the fe ratio. Their contention based on their modelling efforts is that the fe ratio must necessarily be very much lower than the f ratio. After presenting their model results in a few conceptually different ways, they come back to this old standard with a calculated fe ratio of 0.03 (p. 8), an order of magnitude lower than the calculated f ratio. This implies that Fe is recycled in this area with a remarkable efficiency that is unprecedented for any other element ever considered in any other study.

However, there are other literature estimates of fe ratios from several in situ iron budget process studies (some of them cited here) that uniformly estimated higher fe ratios of 0.1-0.5, in other words very similar to typical f ratios for nitrate. The model presented here relies on a multi-year set of measurements of nitrate uptake and incorporation into PON, while several of these prior studies have used limited duration process cruises to attempt much more exhaustive characterizations of the iron cycle, including measurements of iron speciation and photochemistry, iron pool and turnover measurements by the full range of autotrophic and heterotrophic plankton, and Fe export measurements using trace metal clean traps, etc. I'm not saying that these prior studies were right and this one is wrong- after all, we all know that every different measurement method yields a different estimate for even the f ratio of nitrate. This big discrepancy between their very low estimate and previously higher calculated/measured values in other studies does need to be addressed and discussed here, though.

One issue that may be worth considering is touched on just briefly on lines 160-161, temperature. The prior iron budget process studies mentioned above have all been in high latitude temperate or (sub)polar regimes, where water temperatures are much lower than in the Eq Pac HNLC region. Do higher temperatures drive higher rates of biological iron recycling, for instance by grazing microzooplankton? Also, exposure to irradiance is obviously different as well, we know that photoreduction of iron/ligand complexes is a major source of bioavailable iron, and this process is likely far more important under the equatorial sun. Is photochemical recycling of Fe also faster in the tropics? These environmental differences and others such as fundamental differences in biological communities (few or no cyanobacteria in the polar studies, for instance) make me very wary of generalizing estimates obtained for this tropical HNLC region to high latitude HNLC areas, such as is done in the concluding paragraph on p. 10-11.

Specific comments:

Line 53- Can light limitation be neglected when nitrate drawdown is assumed to begin at the source depth of upwelled water, 100-300 days before it reaches the surface (Fig. 4)? Resident deep populations of phytoplankton will require a high Fe/N uptake ratio due to well documented iron/light co-limitation effects, which will progressively decrease as this water advects towards the surface. It seems that a modulated decrease in Fe/N utilization ratios during upwelling might be

more realistic than using a single value.

Lines 98-105- This calculation is another place where flexible Fe:N and Fe:C ratios would be much preferable to the use of a single literature Fe:C value of 12.3 $\mu\text{mol/mol}$. In upwelling plumes, Fe:N utilization ratios change as Fe is depleted relative to N, resulting in either shifts in the ratios of the existing community to become more Fe-efficient, or shifts to more Fe-efficient taxa. A nice illustration of this is Fig. 3 in Bruland et al. 2001, L&O 46 (which however neglects iron recycling completely!). I realize that using flexible drawdown ratios will not change their major conclusion, but it does allow available new iron to be stretched out much further to consume the nitrate before recycling needs to be invoked, and may yield Fe ratios that will more closely resemble those seen in previous studies.

Lines 133-135- I don't agree with their characterization of the Boyd et al. 2012 GRL paper (reference 41 here) as showing that recycled iron was equally available to all phytoplankton groups. Just as for N, they found that pico- and nanophytoplankton dominated the uptake of regenerated Fe, accounting for up to 70% of recycled Fe uptake, even during a temperate spring diatom bloom where diatoms were doing 60% of the primary production.

Lines 162-163- 'other processes' probably needs a little more in depth characterization. For instance, what role does organic complexation play in extending the residence time of Fe in the euphotic zone, relative to N? This major facet of Fe biogeochemistry- and key difference from the N cycle- is barely considered here.

Same section- Scavenging is treated in a fairly dismissive fashion here as well, when in fact it is another one of the biggest differences in the chemistry of Fe and N, and needs to be carefully considered relative to recycling and residence time. It obviously works to minimize the recycling efficiency of Fe compared to N, since Fe is continually being stripped out by passive adsorption onto sinking particles, in addition to biological assimilation. This is supported by the preferential export of Fe relative to nutrient elements documented in the Twining et al. 2014 paper referenced here (Ref 60)- in short, the idea is that while C and nutrients are exponentially remineralized from sinking particles with depth, the scavenging behavior of Fe means it stays preferentially associated with the sinking, shrinking particles, in essence 'distilling' the Fe and driving up Fe:C and Fe:N ratios. Boyd et al. 2017 have a new paper further elaborating on this issue, as well as others that are relevant to this manuscript such as Fe vs f ratios, just out in Nature Geoscience. I realize that this Boyd paper was not available when they wrote and submitted this manuscript, but it would seem like a very good idea to include some discussion of these caveats and inconsistencies between the studies summarized there, in this paper during revisions.

Lines 212-213- It is not at all surprising that doubling dust fluxes doesn't really affect Fe cycling in their model, since as they note earlier dust only supplies less than 1% of new iron entering the Equatorial Pacific. The real issue is variations in the supply of upwelled iron, which they do consider in depth. I'm not sure that this dust calculation adds much to the discussion.

Lines 219-230- Again, I question whether it is appropriate that the major conclusion of a paper dealing with the Equatorial Pacific focuses instead exclusively on the Southern Ocean HNLC area. They are very different regimes, and such broad extrapolations should perhaps be made more cautiously.

Lines 461-462. Don't the measured dissolved Fe concentrations of ~ 0.02 nM mentioned here also include both new and recycled Fe, just like your modeled concentrations? Why then is the value in the model several times higher at 0.07 nM or so? How would it modify your conclusions if you forced dissolved Fe in the model to more closely resemble the lower real measured values in this regime?

Reviewer #2:

Remarks to the Author:

General Comments:

The manuscript, "Recycled iron fuels new primary production in the Equatorial Pacific" is a novel study that deserves publication in Nature Communications. The major claims of the paper are validated through careful and thorough data analysis and modeling. It is clear after reading the paper that the nitrate concentration and isotope data predicts much higher rates of consumption, than can be supplied by new iron. Therefore, they conclude that iron recycling must be responsible for higher than expected amounts of nitrate consumption. Iron recycling has been observed previously, and has been needed to explain primary production rates, but here they have again validated this argument using the stable isotopes of nitrate to show the necessity for iron recycling.

The major thought I was left with after reading the manuscript was: what is the actual mechanism for iron recycling? And why does longer residence time and less upwelling allow for more recycling? I would like to see a more description of how mechanistically iron recycling occurs in the surface ocean. Secondly, I had many questions on the use of the term "available iron" throughout the manuscript. If the amount available were much larger than they assume, would the longer residence time allow greater nitrate consumption? Although not being resupplied with new iron this would mean that organisms are able to continue to access more available iron, and does not necessarily require recycled iron? Which lead me to the question: what pools of available iron are considered recycled?

In terms of the model I had a couple question on the sensitivity to certain parameters. First, what would the affect of a changing C:N ratio have on the model conclusions? By changing the C:N it would create another mechanism for changing the Fe:N ratio, than just by the Fe:C ratio. Secondly, what is the range of uptake measurements using a Rayleigh plot, assuming a closed system, and how good is this closed system approximation? The description of these calculations was in the supplementary material, and I would have liked to see some of the detail in the main text. Additionally, in the supplementary it would have been nice to see a figure that represents the data in Figure 1C, but as a Rayleigh plot. What of the assumptions made would lead to a higher estimate for nitrate uptake than expected?

Overall these comments are to help the authors find the areas where readers may still be skeptical about the results. By addressing the above questions and bulking up the explanation it certain areas, it will make a stronger and more convincing article.

Specific Comments:

Line 64: As mentioned above, would it make sense to plot the data in Rayleigh form?

Line 68: Could you report the isotope effect used here in the main text rather than just the supplementary?

Line 79: Could you report the average isotope values here, as well?

Line 100: How does changing the C:N ratio affect the model?

Lines 105 -106: As mentioned above, maybe you could elaborate here on possible other sources of iron.

Line 146 - 147: Could some of the details of the model be moved into the main text here.

Lines 227-230: Why does a longer residence time only mean more time for iron recycling? Could it also mean more time for nitrate to be removed through other processes, like access to new/different iron pools?

Line 315, 322, 365: One of the "i's" is not capitalized.

Reviewer #3:

Remarks to the Author:

General Comments

The manuscript is really hard work, to a large extent that so much necessary material has been put in the supplementary. Bring tables into the main section and leave only material in SI that is required to dig into the base for some of the conclusions. Having an SI section with its own reference section is just too much to ask a reader.

The model is seriously flawed in allowing only diatoms to take up nitrate and only other phytoplankton to take up ammonium. The authors may wish to look up an existing biogeochemical model for the equator, Dugdale et al. 2002 and Chai et al., 2003. This model uses N as currency and useful rates. It might be helpful to notice that the diatom productivity on the equator is Si limited, so things change with varying silicate along the equator.

It would be helpful if there were a review of previous estimates of the f-fe ratio as defined by others, Boyd? I didn't find a single calculation of the Fe:NO₃ ratio in the upwelling source water. The Fe measurements by Measures are probably the best available and are made on the same water as the nitrate measurements.

Finally, I suggest that the 15N tracer uptakes made on the 2004 and 2005 cruises may be a better estimate of nitrate decline than the use of vertical profiles.

Details:

Figures

Fig. 1. The nitrate concentrations shown in A and B are no greater than about 10, but the concentrations on the x axes in C and D are up to 30, so make it more clear that the data points are from surface to 200m. This brings up another problem, i.e. the difference between source and surface nitrates are used to estimate a consumption (not a rate). The surface water is not directly related to the source water. These water masses are flowing in roughly opposite directions with a huge shear between them. There is no way to get a rate from these numbers at a single station and not even a valid estimate of a delta nitrate.

But now that I read the Legend for Table S1, the source nitrates are calculated from the isotope measurements, which you then plot as isotopes vs the nitrates you have just calculated from the isotopes. Only the last nitrate in the table was measured in what was likely the UC. If a profile exists for these stations, better to use estimates from a density level near the top of the UC, I fail to see how these source nitrates can be used to calculate differences!!

No methods section—(1 paragraph in SI) have to go back and forth between SI and paper. Isotope methods? Analytical chemistry methods? What ships and cruises were made?

No Acknowledgments?

Fig. 2 A. As I said above, I don't trust these estimates, but with a single number in Table S1, how can you show error bars? Where does that come from?

"Scenario 0 uses iron supply and diatom Fe:C requirements that match local observations 16,28. I'm supposed to look these up?? Then Table S2, "Scenario 0 uses observed values at 0 N, 110W 9, 15, doesn't match!

Table S2. Nightmare

!s For Scenario 0 e.g. I see the observations are from 0 n, 110W and I have to go to the legend to find a reference. The reference should be in a column.

Scenario 1 Double dust flux, where did dust flux come from? So I have to look up ref 22 from the legend.

Table S2 Legend ISentence beginnigin " the last two calculations ---- to the end of the legend, I can't make anything out of it.

Table S2 Legend continued: explanation of Fe:C ratios-----used to calculate Fe:N , so what actual Fe:N ratio was used? The use of Redfield to calculate Fe:N is risky, Redfield ratios vary all over the map especially under nutrient limitation. So you can get any nitrate consumption due to varying Redfield. There are publications on that problem. See Figs. 2B, low Fe:C does pretty well!

Are there no published values of Fe:N?

Recommend: Tables should be with the main text
Needs to be a table of applicable Fe measurements
Fig. 3 Conceptual Models

There is a major problem here, in A, shows low uptake of regenerated N (which would be mostly ammonium) by diatoms. In fact diatoms will take up ammonium also, the proportion relative to small phytoplankton depends on concentration of substrate and biomass of both. So by setting diatom uptake of regenerated N, you have set up the conclusion of a preferential remineralization of Fe relative to N. I don't agree with this for two reasons, 1) ammonium is immediately released upon grazing and I can't see how Fe could be released faster. 2) an existing model of the equatorial pacific predicts ammonium uptake of diatoms at 50% of small phytoplankton uptake of ammonium (Chai et al 2002; Dugdale et al., 2003).

Fig. 4A Blue and green dotted lines in middle are biomass?

"Asterisks mark .1%-----targets-----". I don't follow this, which are .1, which are 100%

Fig. S3 Major flaw with no diatom uptake of regenerated N. Diatoms are major players but do not depend on only on nitrate. Further at low silicate, the picoplankton are taking up more nitrate than diatoms, and your model has no provision for nitrate uptake by non-diatoms. So again this will bias your calculations on Fe regeneration.

Main Text:

L 76-77 Statement only applies to specific area of study, large variability along equator.

L93 Tables need to be in main text

L100 Need some measured Fe:N, not Redfield.

L137-138 I can't see that by looking at the figure

L146-148 Problem with references to SI figs and tables, very difficult reading

This page (7) has 4 references to SI

L150-153 Reader can't follow this without some detailed explanation

L159 How do you get a iron recycling rate from nitrate consumption that has no time base, not a rate

168 what are published values of fe-ratio

L170 Where do I find the modeled fe-ratio?

L173 `Or the model construction or parameters are not correct

L185 delete of before iron

L209 I can't see that in Fig. S2

General Comments

The manuscript is really hard work, to a large extent that so much necessary material has been put in the supplementary. Bring tables into the main section and leave only material in SI that is required to dig into the base for some of the conclusions. Having an SI section with its own reference section is just too much to ask a reader.

The model is seriously flawed in allowing only diatoms to take up nitrate and only other phytoplankton to take up ammonium. The authors may wish to look up an existing biogeochemical model for the equator, Dugdale et al. 2002 and Chai et al., 2003. This model uses N as currency and useful rates. It might be helpful to notice that the diatom productivity on the equator is Si limited, so things change with varying silicate along the equator.

It would be helpful if there were a review of previous estimates of the f-fe ratio as defined by others, Boyd? I didn't find a single calculation of the Fe:NO₃ ratio in the upwelling source water. The Fe measurements by Measures are probably the best available and are made on the same water as the nitrate measurements.

Finally, I suggest that the 15N tracer uptakes made on the 2004 and 2005 cruises may be a better estimate of nitrate decline than the use of vertical profiles.

Response to reviews of NCOMMS-17-03587,

“Recycled iron fuels new primary production in the equatorial Pacific Ocean”

By Rafter, Sigman, and Mackey

We appreciate this opportunity to respond to the thoughtful comments on our manuscript. Our revised manuscript has been substantially modified / improved, including the addition of new text and the movement of text, figures, and tables from the supplements to the main text. Below we address all of the Reviewers' comments and detail how these comments influenced the revised manuscript. The format of this response begins with a **Summary of Reviewer's comment** followed by our response and action taken.

REVIEWER 1

Comment 1.1: The calculated fe-ratio is an order of magnitude lower than that estimated in published studies of higher latitude waters, which could be driven by very different environmental conditions. For example: (1) higher temperature at the equator (via increased microzooplankton grazing), (2) stronger light at the equator (increasing photoreduction of iron/ligand complexes and photochemical cycling), (3) larger picophytoplankton presence at the equator.

This is an excellent comment and we appreciate that the Reviewer suggested a concise way to incorporate this discussion into the manuscript. Our new discussion features a more detailed comparison of the high and low latitude fe-ratio that uses much of the Reviewer's wording from the comment to highlight the potential for environmental and biological factors to play an important role in determining the efficiency of iron recycling in these different regions. See text beginning LINE 332:

Comment 1.2: These spatial differences make me wary of generalizing estimates from the tropics to the high latitude HNLC areas for concluding paragraph.

We agree and stated on lines 160-161 (original manuscript) that the estimated rates of iron recycling should not be applied to higher latitude HNLC waters because of temperature differences.

However, we also think it is important to identify the implications of these results. To accommodate the Reviewer's comment, we have removed the final sentence of the Abstract, shortened the 1 paragraph discussion of glacial-interglacial nitrate consumption in the Southern Ocean HNLC, and have added a concluding paragraph (beginning LINE: 406). We think these modifications de-emphasize this discussion without removing it altogether.

Comments 1.3 & 1.4: Authors should consider light limitation affects on Fe/N uptake ratio in model. The Fe/N ratio also shifts as iron becomes limiting because of different taxa.

All calculations of diatom N and Fe consumption are based on observations by Twining et al. (2011) at 20-25 m depth, converted to Fe/N requirements using Redfield C:N of 106:16. We understand that light limiting conditions require additional cellular iron to be allocated to the photosynthetic machinery, raising the Fe/N (Sunda and Huntsman 1997). In contrast, changes in community composition (e.g., Bruland 2001) can make the community more efficient at consuming iron, lowering Fe/N.

In the original manuscript, we explored the potential effects of this Fe/N flexibility by conducting a sensitivity analysis with varying iron uptake efficiencies in Table 1 and Fig. 2B. What we show (and what the reviewer correctly points out) is that this more efficient uptake does not change our conclusions.

The argument then becomes, should variable Fe/N be used in the model, with higher Fe/N at depth (different for each taxa) and lower Fe/N as the community evolves over time. This approach, while more conceptually realistic, would introduce high levels of uncertainty due to the poor agreement between different methods of measuring cellular Fe (see King 2012). Given that the goal of this simple model is to generate conservative estimates for Fe recycling, we have given careful consideration to the relative benefits and drawbacks of each parameter in terms of striking a balance between including the correct level of parameterization on the one hand, and minimizing uncertainty from over-parameterization on the other. In light of this tradeoff, we think that the most conservative approach is to use a fixed Fe/N for each taxa in our model, while adding discussion describing how the results might differ with varying Fe/N in the figures and text. Accordingly, we have adjusted the text (beginning on LINE: 247) to indicate that because of a potentially higher Fe demand at depth, our fixed Fe/N estimates represent conservative values. Furthermore, we have added text discussing how changes in community composition could affect our calculations.

Comment 1.5: Boyd et al., (2012) did not indicate equal availability of recycled iron.

The Reviewer is correct. We removed this reference.

Comment 1.6: More details on the following sentence: “Model estimates are expected to be a lower bound because the model neglects iron loss via scavenging / adhesion to sinking particles and other processes.” For instance, what role does organic complexation play in extending the residence time of Fe in the euphotic zone, relative to N? This major facet of Fe biogeochemistry- and key difference from the N cycle-is barely considered here.

While the specific mechanisms of iron recycling in equatorial Pacific surface ocean are not known and beyond the scope of this manuscript, we have updated the manuscript to consider several important processes and their impact on iron recycling (and our estimates) beginning on Line 332. Additional text responding to this comment is below.

Comment 1.7: Scavenging is dismissed, despite its importance to availability of iron in the ocean.

We considered scavenging very carefully, to the extent that we discussed this with Mick Follows at MIT (an expert in modeling iron chemistry). With his help, we performed some simple calculations to see whether it was reasonable to neglect scavenging in our modeling experiments. With Mick's help we performed several calculations that show that the uptake rate exceeds other loss processes such as scavenging by 7 orders of magnitude (this text is in the Supplementary Information). It follows that parameterizing the many processes that work against iron scavenging in the euphotic zone (e.g., ligands) is unnecessary and would introduce additional unwarranted uncertainty into our simple, yet conservative estimates.

That said, we have moved some discussion of iron scavenging and our modeling parameterization from the Supplement to the Main Text (beginning LINE: 297). A full discussion of the calculations mentioned above are within the Supplementary section titled, "Comparative calculations of iron fluxes in the eastern equatorial Pacific".

Comment 1.8: Not surprising that Last Glacial Maximum dust flux does not have large influence and does not add much to conversation.

We understand why the Reviewer might think this is not surprising, but we feel this brief discussion will benefit readers who are less familiar with atmospheric deposition patterns in this region. For example, just last year there were 2 high profile paleoceanographic papers discussing this very question (Costa et al. in Nature and Winckler et al. in PNAS). Neither of those studies performed this calculation, and we think it is a benefit to the wider oceanographic community to provide it here.

Comment 1.9: Major conclusion of paper is focused on Southern Ocean and not the equatorial Pacific.

We agree with the reviewer that the major conclusion of the paper should focus on the equatorial Pacific and not the Southern Ocean and we see how this may have been confusing in the original version, where the last paragraph discusses the Southern Ocean. However, this paragraph was meant to detail a potentially important *implication* of our findings assuming similar processes occur in other HNLC regions—an insight that most readers are unlikely to make on their own.

In the revised version, we have taken steps to clarify that the potential implications for the Southern Ocean. We have done this by following the formatting used in the seminal Coale et al. 1996 paper that discussed the first successful iron fertilization experiments in the equatorial Pacific, in which the impact of iron fertilization on the Southern Ocean is discussed at the end of the manuscript.

Furthermore, we have made structural changes to the paper, in that we have added substantial Supplementary Material to the main text and have added a summary

paragraph highlighting the study's main findings at the very end of the manuscript. Therefore, the manuscript no longer ends with the Southern Ocean paragraph, and the discussion of the Southern Ocean is a relatively smaller proportion of the overall text. We feel that the revised version does a better job of exploring this potentially important implication of our work, while also avoiding presenting it as a conclusion of the work we present.

Comment 1.10: Why does modeled initial & recycled iron about 0.07 nM when observations are 0.02 nM?

The initial dissolved iron is 0.09 nM and the “new & recycled iron” in Fig. 4 describes the Fe fueling all primary production, which is considerably higher than the observed dissolved Fe because of iron recycling. This comment indicates that we were unclear in describing this aspect of the model and we have adjusted the text accordingly (LINE: 799).

REVIEWER 2

Comment 2.1: Major thought after reading manuscript is: What is the actual mechanism for iron recycling? Would like to see more mechanistic description of iron recycling in the surface ocean.

There are a few processes that are known or are believed to be important for iron recycling, including the photoreduction of organic iron complexes in the surface layer, the presence of iron-binding ligands and grazing, which appears to both increase the solubility of digested iron (Barbeau 1996) and release additional iron-binding ligands (Sato 2007). Text describing these processes was added to the manuscript beginning LINE: 300.

Comment 2.2: Why does longer residence time and less upwelling allow for more recycling?

Linking iron recycling to temporal variability of nitrate consumption is a key observation of our manuscript and we have updated the text to better explain this finding. In short, the current explanation for variable surface nitrate concentrations and nitrate consumption with upwelling is that there is a change in the subsurface source water Fe:Nitrate composition. We argue that this hypothesis is not supported by our new findings and instead suggest that given a fixed rate of iron recycling, you would expect more nitrate consumption with weaker upwelling, which produces a longer residence time. We have created a specific section to more clearly state this mechanism beginning LINE: 377.

Comment 2.3: If the “available iron” were larger than assumed, wouldn't this allow for greater nitrate consumption with increased residence time? If so, this might not require recycled iron.

The Reviewer is correct, and this is why we explicitly examined all potential sources of iron to phytoplankton in the eastern equatorial Pacific with calculations in Table 1 and the Supplementary Information. These include:

Table 1 (0): observed dissolved iron + dust-borne iron with observed solubility;
Table 1 (1): Double dust-borne iron supply;
Table 1 (2): Order of magnitude larger solubility of dust-borne iron
Table 1 (3): dissolved iron concentrations from 200 m
Table 1 (4): Lowest observed Fe:C requirements
NEW Table 1 (5): All particulate iron is bio-available
NEW Table 1 (6): Dissolved iron concentrations from 3000 km west.

We conducted these calculations (many of which are extreme examples) to show that iron cannot explain the nitrate consumption. We have moved calculations 5 and 6 from the supplemental to the main text so that all the sources can be discussed in one location in the manuscript. We believe that this adjustment (compiling the different “available iron” calculations in one place addresses the comment that the Reviewer raised.

Comment 2.4: What pools of available iron are considered recycled?

We have detailed in the updated manuscript that model “recycled” iron is iron regenerated from the initial, upwelled $0.09 \text{ nmol kg}^{-1}$ (from the EUC). See text beginning on LINE: 281.

Comment 2.5: What would the affect of a changing C:N ratio have on the model conclusions?

The Reviewer is correct that variable C:N ratio could influence our results, but species-specific phytoplankton C:N measurements for this region are limited. However, those that are available agree reasonably well with Redfield. For example, a community C:N of ≈ 6 was found by both Coale (1996) (from changes in DIC and nitrate) and by Buesseler (1995) (using sediment flux), which is similar to Redfield C:N of 6.6.

It is nevertheless a valid concern, given that diatoms (the primary consumer of nitrate in this system) can have a variable C:N. To address this comment and acknowledge this uncertainty, we have added error bars to our “potential nitrate consumption” calculations in Fig. 2 that scale with C:N variability of ± 2 from Redfield, which is larger than the 25 to 75 percentile range of C:N values estimated by Martiny (2013 in GBC) for the Pacific Ocean.

Comment 2.6: What is the range of uptake measurements using a Rayleigh plot, assuming a closed system, and how good is this closed system approximation? Would have liked to see this description in the main text and a figure in the SI.

We have added moved text from the SI describing how we solve the Rayleigh equation for the initial subsurface source nitrate concentrations (beginning LINE: 76). We also point to a study (Rafter and Sigman 2016) that gives a necessarily thorough discussion of this approach, which includes figures in “Rayleigh space” (nitrate isotope vs. $\ln(\text{nitrate concentration})$). We are happy to add this figure if the Editor thinks its addition is not excessive.

Comment 2.7: What of the assumptions made (re: closed system behavior) would lead to a higher estimate for nitrate uptake than expected?

Good comment. If this assumption is wrong and new nutrients can resupply the euphotic zone, the estimated nitrate consumption would be an underestimation—exacerbating the difference between “potential” and observed. This is because the “final” nitrate concentration (at the surface) actually represents the drawdown from a much larger amount of nitrate. Furthermore, this would not change the iron-to-nitrate ratio of the upwelled water and cannot account for the apparent iron deficit.

To take into consideration this possible bias, we have adjusted the text (LINE: 461) to indicate that our estimated nitrate consumption is a conservative value if the assumption of a closed system is violated.

Specific Reviewer 2 Comments:

Rayleigh Plot? Discussed above.

Report the isotope effect? Discussed above.

Changing C:N: Discussed above.

Possible other sources of iron: Discussed above.

Move some details of the model to main text: Much of the model details have been moved to the main text.

What about other pools of iron? Discussed above.

One of the “I’s” is not capitalized. Fixed.

REVIEWER 3

“General Comments”

Comment 3.1: Bring Tables into the Main Text.

We have brought these into the Main Text.

Comment 3.2: Having an SI section with its own reference section is just too much to ask a reader.

We understand the Reviewer’s concern about the length of the supplement; however, as publications are increasingly moving toward online publication, it is also becoming increasingly common for longer supplemental sections to be included and linked directly from the main manuscript. This offers the advantage of allowing authors to include higher levels of detail about their data, analyses, modeling assumptions and equations, and rationales (including references that were considered) than was ever possible before. It also helps authors conform to the new Data Management guidelines from NSF, which require raw data to be made publically available – often this level of reporting is too dense for inclusion in the main text of an article, but it can be easily included in a supplemental section.

We feel that the thorough detail we provide in the supplement is useful for several reasons. First, this interdisciplinary study will likely be read by people from a range of disciplines, so providing the relevant background assumptions in a complete and systematic way will facilitate broader understanding of the methods and

conclusions across different fields of oceanography. Second, we provide complete and transparent access to the model, including all equations and input variable assumptions (along with the references from which they were gathered), such that our work can easily be replicated and tested by other researchers. Finally, we feel that the online structure of Nature Communications, which allows readers to easily access the supplement by clicking on a link in real time, will not overburden readers who chose to do so.

Comment 3.3: The model is seriously flawed in allowing only diatoms to take up nitrate and only other phytoplankton to take up ammonium.

While we appreciate the Reviewer's concern that the N uptake assumptions are an oversimplification, there are several reasons why representing the system this way is valid.

First, with respect to diatom N requirements, several studies find that "diatoms at times take up the major portion of the nitrate." (Dugdale 2011 in DSR; about the eastern equatorial Pacific).

Second, we understand that dinoflagellates also consume some nitrate in the eastern equatorial Pacific (see Parker (2011)). We address this in our conceptual model (Fig. 3) and wrote the following Supplementary Text in the original manuscript:

"Dinoflagellates may also consume nitrate at this site ¹², but because of their higher Fe:C requirements (14.2 $\mu\text{mol mol}^{-1}$ ⁹), their contribution to nitrate consumption would only serve to increase the required rate of iron recycling."

To address this comment, we have moved the above text from the Supplement to the Main Text (LINE: 236).

Third, while diatoms and dinoflagellates can take up ammonium, the higher surface area to volume ratio of picoplankton gives this group a distinct advantage in taking up this recycled resource. In this system where nitrate is abundant, the pressure for large and small cells to compete for ammonium would also be much lower than in an N limited system.

Finally, the relative rates of nitrate and ammonium uptake by competing picoplankton and diatoms is poorly constrained, and adding this layer of complexity to the model would therefore introduce unnecessary uncertainty to our estimates. We therefore used the simplest assumptions based on points above when building the model.

We stress that the basis of a simple model like ours is not to parameterize every pathway and reservoir in explicit detail (nor would this be possible as many of the fluxes are unknown or poorly constrained), but rather to articulate the most influential pathways and test potential implications. We feel that given the tradeoff

between adding additional parameters to increase realism and the additional uncertainty this necessarily introduces, our simple model strikes an appropriate balance in the set of parameters and assumptions we have applied.

Comment 3.4: It might be helpful to notice that the diatom productivity on the equator is Si limited, so things change with varying silicate along the equator.

The majority view of equatorial Pacific biogeochemistry is that diatom growth rates and (most importantly for our study) nitrate uptake respond to the addition of iron and not silicic acid (e.g. Mark Brzezinski 2008 and 2011 and Coale et al., 1996). Brzezinski et al. (2011) explicitly states that “the main effect of Si (addition) was to regulate diatom silicification”. Given this understanding, we do not expect variability of silicate to influence nitrate consumption and discuss this on LINE: 295.

Comment 3.5: Review of previous fe-ratio.

To accommodate this and the comment above we compare fe-ratios in the revised manuscript (see comment above).

Comment 3.6: I didn't find a single calculation of the Fe:NO₃ ratio in the upwelling source water. The Fe measurements by Measures are probably the best available and are made on the same waters as the nitrate measurements.

The beginning of this manuscript focuses on the source of upwelling waters and its geochemistry. This includes identifying source water nitrate concentrations (from the EqPac Biocomplexity cruises and our own) and the Measures Fe data (published by his student Kaupp in 2011). We updated the text (LINE: 835) to give the Fe:NO₃ ratio for Scenario 0.

Comment 3.7: I suggest that the 15N tracer uptakes made on the 2004 and 2005 cruises may be a better estimate of nitrate decline than the use of vertical profiles.

We actually use the Parker et al. measurements of nitrate uptake *rates* as a test for our model and find a close correspondence (beginning on LINE: 312). What these measurements do not provide is a total consumed nitrate concentration. Nor do they allow us to identify source waters or constrain the method of nutrient delivery (closed vs. open “chemostat” system nutrient supply). In contrast, the vertical profiles of nitrate isotopes and concentrations provide excellent constraints on source water geochemistry and nutrient delivery (as discussed here and more thoroughly in Rafter and Sigman 2016 in L&O).

Comment 3.8: Nitrate concentrations scale is inconsistent between map and vertical profiles.

The Reviewer is correct that the scales are different, which allows differences in the figures to be easier for the reader to interpret. We have also edited the caption to state that the scales are different and state that “Moving from right to left—higher to lower nitrate concentrations—coincides with sampling depths from 200 m to the surface.”

“Figures”

Comment 3.9: The surface water is not directly related to the source water. There is no way to get a rate from these numbers at a single station and not even a valid estimate of a delta nitrate (consumption).

As is detailed in the current manuscript and prior work (Rafter and Sigman 2016 in L&O), the nitrate concentrations and isotopes predict that the source of equatorial Pacific surface nitrate is Equatorial Under Current (EUC) waters directly below. This occurs even though the mean surface circulation is counter to the EUC. Furthermore, lateral advection of surface nitrate is minimal because Ekman divergence at the equator moves waters both westward and poleward. These considerations are discussed in detail in our L&O paper, and discussing them again in this new manuscript would be both redundant and beyond the scope of this study.

Comment 3.10: Better to use estimates from a density level near the top of the UC, I fail to see how these source nitrates can be used to calculate differences!!

The methods we use to identify subsurface source waters point to the shallower EUC density levels as the source of upwelled waters in the eastern equatorial Pacific and we therefore use these values in the calculations. It follows that the difference between initial and final values represents consumption.

Comment 3.11: No methods section? No acknowledgments?

These sections are available in both the Main Text and Supplementary Material sections.

Comment 3.12: As I said above, I don't trust these estimates, but with a single number in Table S1, how can you show error bars? Where does that come from?

We have adjusted the manuscript to identify the error bar values. The estimated nitrate consumption values represent observed EUC (the source water) nitrate concentration variability. The “potential” nitrate consumption error bars reflect possible C:N variability (addressing the comments of this and other Reviewers).

Comment 3.13: Comments about source water geochemistry referencing in Table S2.

We apologize for not being clear about the parameters going into these calculations, but we thought it was clear that “available iron” in Table 2 referred to the iron used for these calculations. We have adjusted the text to clarify this and have added the references to the final column of the Table (as requested).

Comment 3.14: Table S2 Legend. I can't make anything out of it.

We have adjusted the manuscript to clarify this text.

Comment 3.15: Using Redfield to calculate Fe:C and Fe:N is risky. Are there no published Fe:N?

We agree, and this is why we used Fe requirements observed at the same station location during the EqPaci Biocomplexity cruise (Twining et al. 2011). We further take into account Redfield C:N variability in our error calculations (see above).

Comment 3.16: Tables should be with the main text.

We agree and have moved the tables to the main text.

Comment 3.17: Conceptual models show low uptake of regenerated N... So by setting diatom uptake of regenerated N, you have set up the conclusion of a preferential remineralization of Fe relative to N.

As we discuss above, it is reasonable to assume that diatom N quotas are met by nitrate given that studies in this region have observed this process, and given that nitrate is in abundance. Furthermore, regardless of which taxa are consuming nitrate, there is no Fe physiological requirement that would allow for the complete consumption of nitrate without recycled Fe. This is the point of our study and is well detailed throughout with different possible Fe sources / Fe:N requirements (the “scenarios”), calculations, and sensitivity analyses. We have tried to make the assumptions and caveats of the model as transparent as possible by detailing them in the text and supplement, and by moving important parts of the supplement into the main text as the Reviewer suggested.

Comment 3.18: Ammonium is immediately released upon grazing and I can't see how Fe could be released faster.

To be clear, a major implication of our work is that iron is preferentially retained relative to nitrogen in the system, predicting that sinking Fe:N will be lower than the community Fe:N requirements. This does not make any specific predictions about the “release” of N and Fe during grazing, although we describe some of these processes based on this and other Reviewers' comments. For example, studies show that processing in the acidic guts of protozoan grazers increases the bioavailability of iron. This and other potential mechanisms for iron recycling are discussed included in the updated manuscript (see above).

Comment 3.19: Other models predict ammonium uptake of diatoms at 50% of small phytoplankton uptake of ammonium.

Please see our responses above on why we have made these assumptions and how we transparently present the caveats.

Comment 3.20: Fig. 4A. Blue and green dotted lines in middle are biomass? Asterisks are confusing.

As is shown to the right of those lines, these represent “biomass N [$\mu\text{mol kg}^{-1}$]”. We have modified the text to clarify that we tuned maximum phytoplankton growth rates to match observed biomass at the 0.1 and 100% light levels so that these aspects of the figure are more clear.

Comment 3.21: Fig. S3 Major flaw with no diatom uptake of regenerated N.... So again this will bias your calculations on Fe regeneration.

Please see our responses above.

“Main Text”

Comment 3.22: Statement of source water nitrate concentrations only applies to specific area of study, large variability along equator.

This is correct and this source water nitrate concentration spatial variability is detailed in the 2016 L&O paper discussed above (fittingly titled, “Spatial distribution and temporal variation of nitrate nitrogen and oxygen isotopes in the upper equatorial Pacific Ocean”).

In the present study we are only observing a very narrow range of the equator— 0°N , 110° —where 5 years of observations show relatively little source water nitrate variability (either concentration or isotopically). Focusing on this well-constrained region is one factor that allows us to make use of a simple conceptual model.

Comment 3.23: Tables need to be in main text.

We have modified the text to include the table.

Comment 3.24: Need some measured Fe:N, not Redfield.

We use a Fe:N physiological requirement that is based on synchrotron x-ray fluorescence observations at our study site by Twining et al., (2011). Their study provides an estimated Fe:C value (they cannot directly measure C) and we convert this to Fe:N using Redfield values. We take into account C:N variability in the updated manuscript (see discussion above and Fig. 2 caption and error bars).

Comment 3.25: I can't see that (necessity for a higher Fe:N remineralization ratio) by looking at the figure.

The Reviewer is correct that the wrong figure was referred to, and we have corrected this in the text.

Comment 3.26: Problem with references to SI... difficult reading.

We have modified the text to reduce the amount of SI material.

Comment 3.27: Reader can't follow this (model sensitivity to Fe recycling) without some detailed explanation.

Good comment. We have adjusted the text and moved SI figure 1 to the main text to clarify this description (see above).

Comment 3.28: How do you get an iron recycling rate from nitrate consumption that has no time base, not a rate?

Our study has three parts: (1) observations of nitrate consumption at 0°N , 110°W , (2) simple stoichiometric-based calculations of Fe requirements from this site, and (3) simple box model to identify necessary rates of elemental cycling. The rates are derived from the simple box model.

Comment 3.29: What are published values of fe-ratio?

This is an important comment that has been repeated by several reviewers. We have adjusted the text accordingly (see above).

Comment 3.30: How do I find the modeled fe-ratio?

The text states that the f- and fe-ratios “can be quantified in terms of the uptake ratio of “new” (i.e. upwelled) nutrient to new-plus-recycled nutrient”. This is how we calculated the reported values. They are not directly outputted from the model, but rather calculated from the model outputs.

Comment 3.31: Or the model construction or parameters are not correct (concerning lower fe-ratio to f-ratio).

It appears that this comment wants to insert the above language into the text. We agree that “the model construction and parameters are not correct” because no models are correct, but some are useful. However, regardless of model settings / parameterization, we can say that the fe-ratio must be lower than the f-ratio based only on the stoichiometric calculations. We have adjusted the text to clarify this statement.

Comment 3.32: typo. Fixed.

Comment 3.33: I can't see that (changes in upwelling affecting the surface nitrate pool) in Fig. S2.

This statement and figure caption were not as clear as they could be, so we adjusted and moved the text into the main part of the manuscript.

“General comments”

Same comments as above.

Reviewers' Comments:

Reviewer #1:

Remarks to the Author:

In my opinion, the authors have done a good job of responding to my comments and those of the other reviewers. I was pleased to see that they now recognize physical/chemical/biological differences between the Equatorial Pacific and the high latitude HNLC areas, and have removed speculations about applicability of their findings to the Southern Ocean from the concluding paragraph (the short remaining text on this is fine, as it doesn't stand out as the major point of the paper any more). I'm not sure I believe their Follows-assisted conclusion that scavenging is completely negligible in this regime, as it is based on another modeling effort rather than in situ measurements, but I am willing to accept that it won't change their conclusions drastically. In general, they've done a credible job of supporting their conclusions and putting them into the context of the existing literature.

Reviewer #2:

Remarks to the Author:

General Comments:

The revised manuscript has addressed all issues in the first draft. The article is now clearly written, organized and shows a clear explanation for the necessity for iron recycling to explain nitrate consumption in the equatorial Pacific Ocean. I only had one minor comment described below.

Minor Comment:

Line 304: Repetitive of previous sentence, can be removed.

Reviewer #3:

Remarks to the Author:

I have finished my evaluation of the responses of the authors to my review. They have taken all of them seriously and I believe the result is a much more reader friendly version. They have also considered my scientific reservations adequately and I have no objection to publication in its revised form.